# OpenReview forum: "Simplicity Prevails: Rethinking Negative Preference Optimization for LLM Unlearning"
_NeurIPS.cc/2025/Conference — NeurIPS 2025 poster_

### Official Review · Reviewer_s2gF · 2025-06-08

**Clarity:** 1
**Significance:** 3
**Originality:** 2
**Rating:** 4
**Confidence:** 5

**Summary:**

This paper addresses limitations of an existing loss for LLM unlearning (NPO) and proposes to remove the dependency on a reference model (the model prior to unlearning). The improved loss shows better unlearning vs. utility performance on three LLM unlearning datasets and provides more robustness against relearning attacks.

**Questions:**

Q1) I assume that dropping the dependency on a reference model improves efficiency (as also discussed by SimPO). Does SimDPO improve efficiency over NPO in terms of e.g. runtime or memory-usage?

Q2) What does "ununlearn" mean in Fig1 a)?

*Typo*: gredient -> gradient (L235)

**Ethical Concerns:**

["NO or VERY MINOR ethics concerns only"]

**Final Justification:**

My concerns regarding Weakness 1 have been resolved during the discussion phase. My final assessment is based on the weakness (W2) that the broader impact of this work is rather limited since the scope of this work is only to improve an existing unlearning method (NPO). This concern is also shared with two other reviewers (Reviewer JEx6, Reviewer BfAc). Still, the work appears significant for the unlearning community, thus my increased score and final assessment.

**Limitations:**

Yes, in the Appendix.

**Paper Formatting Concerns:**

No formatting concerns.

**Quality:**

3

**Strengths And Weaknesses:**

**Strengths**
- In general, the paper is in good shape in terms of writing quality (with exceptions highlighted below).
- The paper provides sufficient empirical evidence that SimNPO is better than baselines considering the unlearning-utility trade-off in LLM unlearning.
- The paper also provides interesting findings, in particular provides evidence that SimNPO is more robustness to relearning attacks.

**Weaknesses**

W1) The approach of addressing and overcoming limitations of existing methods in unlearning is generally good. The problem here is that the discussion of the identified limitations in section 4 requires more clarity. In particular:
- Regarding Limitation 1: It is unclear what the term "unlearning power" refers to. Do you mean the gradient is too small when the reference model is highly certain (and vice versa)? This term needs to be clarified in more technical terms, especially because the "unlearning power" and "optimization power" are used later on in the paper as well.
- Regarding Limitation 2: The adaptive weight is close to 1 in the first epoch, and the paper suggests that this has a bad effect on the utility. However, Figure 3 d) shows no change in utility after the first epoch. Maybe studying epoch 2 (when utility starts to drop) would be more effective for this argument. The main problem of this paragraph is, however, that it remains unclear if the reference model is really the underlying issue (as claimed in L220) or rather the missing length-normalization.

W2) The method boils down to a simple fix that has been proposed already in a different context (preference optimization / SimPO), which limits the technical contribution of this paper. Especially the critical length-normalization has also been discussed in the SimPO paper already. While the empirical results are strong and potentially helpful for the unlearning community, the paper would benefit from a stronger discussion on the differences introduced by the unlearning context. For example, the gamma term introduced in SimPO does not seem to be required for unlearning but remains part of the SimNPO loss.

---

> ### Author Rebuttal · Authors · 2025-07-30
>
> We sincerely thank Reviewer s2gF for the thoughtful and constructive feedback. Below, we provide detailed responses to each of the key questions raised.
>
> # 1. Response to the unlearning power or optimization power in Limitation 1
> We apologize for the lack of clarity in our original writing. In this context, both "unlearning power" and "optimization power" refer to the adaptive weight allocated for forgetting data gradients.
>
> This can be seen by examining the adaptive weight used in the NPO gradient (Eq. 3): $w_{\theta}(x, y) = \frac{2 \, \pi_\theta(y \mid x)^\beta}{\pi_\theta(y \mid x)^\beta + \pi_{\text{ref}}(y \mid x)^\beta}$. This weighting scheme causes NPO to assign lower gradient weights to samples with higher output probabilities under the reference model, $\pi_{\text{ref}}(y \mid x)$. However, samples with higher $\pi_{\text{ref}}$ are actually harder to forget than those with lower reference probabilities. As a result, if $\pi_{\text{ref}}$ varies across the forget set in a way that does not accurately reflect the true difficulty of unlearning a sample, this leads to uneven and biased gradient weight allocation during unlearning optimization.
>
> This bias is particularly evident when forget examples vary in memorization level (Example 1) or response length (Example 2). For example, in Example 1, NPO performs poorly on strongly memorized (harder) samples–characterized by higher $\pi_{\text{ref}}$--yielding a forget quality of 0.00, while achieving a forget quality of 0.47 on weakly memorized (easier) samples, as shown in Figure 1(b).
>
> We will revise the manuscript to clarify this terminology and ensure consistency throughout.
>
> # 2. Response to Limitation 2
> Thank you for this thoughtful question. We would like to clarify that the effect of biased gradient weights on utility may not manifest immediately after the first epoch. Unlearning is a progressive process in which the model gradually diverges from the original (pre-unlearning) distribution. As such, **there can be a lag between early biased updates and their downstream impact on utility**, which may become apparent only after a few optimization steps (e.g., around epoch 2 or 3 as noted).
>
> Importantly, the core message in Limitation 2 is why the early unlearning dynamics are suboptimal for utility. As mentioned in Line 223, our analysis suggests that during the early training epochs, NPO’s adaptive weights ($w \approx 1$) are largely insensitive to variations in the difficulty of the forgetting data, such as differences in memorization strength or response length. This leads to uniform unlearning updates, failing to properly prioritize harder-to-forget examples.
>
>
>
> Furthermore, **the reference model is indeed the underlying issue**, as it serves as the mechanism through which unlearning guidance might be unevenly applied across forgetting data of varying difficulty. To elaborate, let us revisit the reference-dependent weighting term in NPO’s gradient (Eq. 3): $w_{\boldsymbol{\theta}}(x, y) = \frac{2 \, \pi_\theta(y \mid x)^\beta}{\pi_\theta(y \mid x)^\beta + \pi_{\text{ref}}(y \mid x)^\beta}$. The influence of the reference model shapes the gradient weights in a way that disproportionately favors easier-to-forget samples (i.e., those with lower output probabilities $\pi_{\text{ref}}(y \mid x)$), thereby introducing bias into the unlearning process, as discussed in Response 1.
>
> As a result, if the reference model $\pi_{\text{ref}}$ varies across the forget set in a way that does not accurately reflect the true difficulty of unlearning a sample, this leads to uneven and biased forgetting.  For example, even when **forgetting data samples have similar response lengths, the reference model bias persists if they differ in memorization levels**. In Example 1, NPO performs poorly on strongly memorized (harder) samples, characterized by higher $\pi_{\text{ref}}$ values, achieving a forget quality of only 0.00. In contrast, it attains a forget quality of 0.47 on weakly memorized (easier) samples, as shown in Figure 1(b).
>
> The above illustrates that the bias induced by the reference model is not merely a function of response length, but is fundamentally tied to how the model prioritizes forgetting across varying difficulty levels.
>
> Lastly, we conducted additional experiments using a modified NPO objective that incorporates length normalization, in order to examine whether it can directly address the reference model bias in NPO. The NPO variant with length normalization (denoted as NPO-LN) is defined as: $\ell_{NPO\text{-}LN}(\theta) = -\frac{2}{\beta} \log \sigma \left( -\frac{\beta}{|y|} \log \left( \frac{\pi_\theta(y \mid x)}{\pi_{\text{ref}}(y \mid x)} \right) \right)$.
>
> Following the setting of $Example 1$, we construct a controlled scenario where samples within the forget set exhibit **different levels of memorization**, where the subset $D_{f,1}$ is strongly memorized and the remaining forget set $D_{f,2}$ is minimally memorized by the reference model.
>
> We then perform unlearning over $D_{f,1} \cup D_{f,2}$ using NPO, NPO-LN, and SimNPO. As shown in **Table R5**, although NPO-LN improves upon the original NPO, it still underperforms SimNPO. This suggests that the reference model issue is only partially mitigated by length normalization, and SimNPO remains more effective in addressing this bias.
>
> Table R5: Comparison across NPO, NPO-LN, and SimNPO on TOFU Forget05.
>
> | **Method**   | **Forget Quality on $D_{f,1}$** | **Forget Quality on $D_{f,2}$** | **Model Utility** |
> |:------------:|:----------------------------------------------:|:----------------------------------------------:|:------------------:|
> | NPO          |                     0.00                       |                    0.47                        |       0.49         |
> | NPO-LN       |                     0.10                       |                    0.55                        |       0.51         |
> | **SimNPO**   |                 **0.70**                       |                  **0.70**                      |     **0.57**       |
>
>
> # 3. Response to novelty and hyperparameter $\gamma$
>
> While SimNPO may appear to be a simple methodological modification to SimPO, we respectfully emphasize that our work represents a novel contribution tailored to the domain of **LLM unlearning**, a substantially different problem space than the context originally considered in SimPO. Specifically, SimNPO was motivated, analyzed, and rigorously validated within the unlearning context: Our work fundamentally rethinks the design of unlearning objectives by identifying and overcoming the reference model bias in NPO.
>
> In particular, we are the first to (1) uncover the disproportionate gradient influence imposed by the reference model across forgetting data of varying difficulty (as exemplified via memorization level and response length), and (2) empirically and theoretically characterize how this bias leads to suboptimal forgetting outcomes. These findings have not been addressed in prior work. Our contributions go beyond methodological substitution. We provide a systematic investigation, including theoretical analysis via a Markov chain mixture model, empirical validation across multiple benchmarks, and robustness studies.
>
> In addition, we kept $\gamma$ in SimNPO's objective for ease of comparison with SimPO. However, we have clearly stated in Line 252 that $\gamma$ is not used (i.e., $\gamma = 0$), and we provided the rationale for this choice in Lines 252–256, explaining why it is not necessary for the SimNPO objective.
>
> # 4. Response to the computation efficiency and term “Ununlearn”
> Thank you for your insightful suggestion. In response, we conducted a comparison of the computational efficiency among Retrain, NPO, and SimNPO. As shown in **Table R6**, NPO introduces additional overhead during the unlearning process for two reasons: (1) it requires storing an extra reference model, which increases GPU memory usage; and (2) it must compute $\log(\pi_{\text{ref}}(y \mid x))$ at each training step, leading to increased computational time.
>
> Table R6: Comparison of GPU memory usage and training time for Retrain, NPO, and SimNPO on the TOFU Forget05.
>
> | Method   | Memory (GB) | Time (min) |
> |:--------:|:-----------:|:----------:|
> | Retrain  |     20      |    120     |
> | NPO      |     27      |     36     |
> | SimNPO   |     21      |     25     |
>
> In addition, we apologize for the confusion caused by the unclear use of the term "ununlearn" in the manuscript. In our context, "ununlearn" refers to the original model before any unlearning is performed.

---

> ### Comment · Reviewer_s2gF · 2025-08-03
> **Response to Rebuttal**
>
> Thank you for the detailed clarifications and suggested edits to the manuscript.
>
> After your clarifications I still disagree with the framing that the output probability of the reference model (model prior to unlearning) reflects the "true sample-specific unlearning difficulty" (L197). I think this might be misleading since "unlearning difficulty" can differ for responses even if their output probability is the same, meaning we do not know the distance to the "unlearn boundary" (which is rather challenging to asses I assume). However, I do think it becomes more clear now that the dependence of the NPO loss on output probabilities of the reference model is the underlying issue.
>
> I would have a follow-up question after your clarifications: Do the performance improvements in the main experiments (Section 6.2) mainly stem from addressing the problem of varying output probabilities for different response lengths? Meaning, are there actually different "memorization levels" (output probabilities) for responses of similar length when applying unlearning to models finetuned on the full dataset (Table 1), or does this only occur in the synthetic experiments introduced in Appendix C? I think knowing this would be helpful to better understand the effects on the final performance improvements.
>
> Furthermore, it would be helpful if you add experimental setups / hyperparameters for the additional experiments in Table R5 and R6, in particular if you fixed a specific $\beta$ or chose the best-performing models.

---

> ### Author Response · Authors · 2025-08-04
> **Thank you and further response**
>
> Thank you for the follow-up comments.
>
> (1) We are glad to hear that the underlying issue of reference model bias is now clearer. We also appreciate your suggestion to reframe and clarify our use of the term “true sample-specific unlearning difficulty” (Line 197) to more accurately capture its relationship with the reference model’s output probabilities. In the revision, we will provide a more precise explanation of this concept by contextualizing “sample-wise difficulty” in terms of how the reference model’s output probability mediates the unlearning power in NPO, leading to varying treatment of forget samples based on their output probability.
>
>
> In addition, we would like to further clarify **our original intent** behind the term “true sample-specific unlearning difficulty.” Our goal was to highlight that NPO’s reliance on $\pi_{\text{ref}}$ can introduce unintended bias when handling diverse forget data. Specifically, the reference model, being fixed and unadjusted for unlearning, may exhibit model-specific, stereotyped confidence patterns across forget examples. These patterns may not accurately reflect the actual difficulty of forgetting each sample. We did **not** intend to suggest that the output probability alone constitutes a sufficient condition for quantifying sample-wise unlearning difficulty, but rather a useful lens for analyzing how and where reference model bias may arise in terms of the reference model’s output probabilities against forget samples.
>
>
> (2)  **Thank you for the additional insightful comment regarding output probability and response length.** We would like to clarify that even when forget examples have similar response lengths, they can still exhibit varying memorization levels, which in turn induce reference model bias through differing output probabilities.
>
>
> This is supported by **PrivLeak** in Figure 1(d) on MUSE-News and Table 2 on both MUSE-News and Books. Specifically, in the MUSE dataset, the forget set $\mathcal{D}_f$ contains **samples with consistent token lengths**. The PrivLeak metric, based on membership inference attacks, serves as a proxy for the memorization indicator, where values closer to zero are better in unlearning (i.e., closer to Retrain). As shown in Table 2, SimNPO achieves much lower PrivLeak than NPO, indicating better unlearning to erase the memorization of forget data.
>
>
> In addition, we chose to conduct the synthetic memorization experiment in Example 1 (i.e., Appendix C) based on TOFU as this represents a direct case of memorization. We explicitly fine-tune the model on one subset of the forget set while leaving another untouched, thereby creating **strongly** and **weakly** memorized samples. This synthetic setup was designed for two reasons: to clearly illustrate the concept of memorization via controlled fine-tuning; to isolate memorization effects from response length (as the splitting of the two subsets is not based on response length). This allows for a cleaner comparison of forgetting performance across memorization levels.
>
>
> We hope the above clarifies the role of memorization beyond response length and reinforces the motivation behind both our main and synthetic experiments.
>
>
> (3)  The detailed setup for Table R5 is as follows:
>
> - Dataset: TOFU Forget05
> - Model: LLaMA-2-Chat 7B
> - Methods: NPO, NPO-LN, SimNPO
> - Evaluation metric: Forget quality and model utility
> - Hyperparameters:
>   - Learning rate: 1e-5
>   - Beta: 0.1 for NPO, 2.5 for SimNPO, 3.5 for NPO-LN
>
>
> Please note that for hyperparameter selection, we performed a grid search based on the ranges used in Appendix I2, and selected the best models with the highest forget quality gain but preserving a reasonable utility performance for each method.
>
> The setup for Table R6 follows Table R5. We use 8×A6000 GPUs to conduct experiments and further measure running time and memory usage. Running time is measured in minutes required for running an unlearning algorithm, and memory refers to peak GPU memory usage during unlearning.

---

> ### Comment · Reviewer_s2gF · 2025-08-04
> **Response to Comment**
>
> Thank you for the additional clarifications, it is certainly more clear now and I agree with your suggestions to improve the usage of the terms "unlearning difficulty" and "optimization power". I still share the concerns also raised by Reviewer JEx6 and Reviewer BfAc regarding the broader impact and I will continue monitoring the discussion with the other reviewers. However, I do acknowledge the clarifications during the rebuttal regarding my other concerns, as well as the relevance of this contribution to the unlearning community. I will increase my score accordingly.

---

> > ### Author Response · Authors · 2025-08-04
> > **Thank You for Raising Your Score and Acknowledging Our Response and Discussion**
> >
> > Thank you very much for agreeing to raise your score. We sincerely appreciate your recognition of the relevance of our contribution to the unlearning community and the clarifications and responses provided during the rebuttal and discussion phases.
> >
> > We will surely revise the use of “unlearning difficulty” and “optimization power” in the paper to make these terminologies clearer and more precise.
> >
> > If possible, we would be grateful if you could share any further details on the broader impact concerns. We clearly checked the comments and ratings from Reviewer JEx6 and Reviewer BfAc, and were not quite clear what the broader impact precisely refers to. We are happy to continue the discussion and address any additional questions.
> >
> > Thank you again for your time and thoughtful feedback.

---

> > > ### Comment · Reviewer_s2gF · 2025-08-05
> > >
> > > Thank you for your request for additional clarification. To clarify, I was referring to the broader impact beyond unlearning as indicated in my review (Weakness 2), but also noted by Reviewer JEx6 (Weakness 1) and Reviewer BfAc (Weakness 1). I share these concerns and will continue monitoring the discussion with the other reviewers. Still, I have considered your responses and acknowledge the contributions to the unlearning community, which is already reflected in my updated score.

---

> > > > ### Author Response · Authors · 2025-08-06
> > > > **Sincere Thanks for the Clarification on Broader Impact and Score Adjustment**
> > > >
> > > > We sincerely appreciate your recognition of our work and your further clarification on the broader impact.
> > > >
> > > > We would like to take this opportunity to reiterate why we believe that articulating NPO’s limitations and introducing the design of SimNPO is meaningful, with the potential to yield a broader impact for the community.
> > > >
> > > > Since NPO has been widely adopted as a state-of-the-art approach in unlearning research, understanding its limitations can provide valuable guidance for future developments in the area of machine unlearning. For instance, its reliance on a reference model may introduce previously unrecognized bias, which could inform the design of future unlearning loss functions that use a reference model to quantify unlearning performance. Moreover, analyzing how different types of forget data (e.g., variations in response length or memorization level) influence unlearning optimization broadens the discussion on forget data variation, a topic also explored in image classification [1].
> > > >
> > > > While SimNPO may appear to be a simple fix inspired by SimPO, it is grounded in careful insights and detailed analysis. For example, our work includes synthetic experiments (Figures 1(b) and 5), convergence analysis (Appendix G), adaptive weight behavior studies (Figures 3 and 4), comprehensive benchmark evaluations (Tables 1, 2, and A4), and robustness comparisons (Figure 6). Although adapting SimPO to unlearning may seem straightforward methodologically, demonstrating why and how this adaptation improves unlearning performance required substantial conceptual, analytical, and experimental effort.
> > > >
> > > > We hope this additional context could further clarify the broader impact from our viewpoint. Thank you again for your kind and constructive feedback.
> > > >
> > > > > [1] Zhao, Kairan, et al. "What makes unlearning hard and what to do about it." Advances in Neural Information Processing Systems 37 (2024): 12293-12333.

---

### Official Review · Reviewer_BfAc · 2025-06-26

**Clarity:** 4
**Significance:** 3
**Originality:** 3
**Rating:** 5
**Confidence:** 3

**Summary:**

This paper focuses on the forgotten performance bias introduced by the  reference model item in the loss function of vanilla negative preference optimization. The authors propose a new LLM unlearning mechanism, SimNPO, to mitigate the reference-model-introduced bias. The authors present details explanations  and validate the effectiveness of SimNPO through three benchmarks. The experimental results support the authors' claims.

**Questions:**

1.Does the  forgotten performance bias in NPO always exist, or does it only occur when there is an over-reliance on the reference model? I am curious about the conditions under which such over-reliance occurs. Is there a clear criterion to identify when this happens?
2. Since SimNPO uses the response length $|y|$ to replace the function of the reference model $\pi_{ref}$ in the loss function, is there a risk of over-reliance on $|y|$, potentially leading to a new bias phenomenon?

**Ethical Concerns:**

["NO or VERY MINOR ethics concerns only"]

**Limitations:**

The authors have discussed the potential limitations and societal impacts in the appendix, which appears to be a rational and thoughtful consideration.

**Paper Formatting Concerns:**

NA. Well-written.

**Quality:**

3

**Strengths And Weaknesses:**

S：
1. The paper is well-written and structured, making it a pleasure to review.
2. The paper thoroughly examines the bias limitations of the NPO method and provides a comprehensive analysis of the advantages of the proposed SimNPO. This solidifies the technical claims and offers valuable insights for the related communities into more fair unlearning mechanisms.

W:
1. The paper focuses primarily on the limitations of a recently published method, NPO, which  might be considered somewhat limited in scope.
2. Since the work is based on the limitations of an existing method, it would be beneficial to analyze the potential negative impacts of removing the reference model in the loss function of SimNPO. For instance, could it negatively affect utility? Even if not, it would be helpful to explain why SimNPO can still achieve higher utility compared to NPO despite removing the reference model.

---

> ### Author Rebuttal · Authors · 2025-07-30
>
> We greatly appreciate Reviewer BfAc’s thoughtful and constructive feedback. Below, we provide detailed responses to each of the key points raised.
>
> # 1. Response to focusing primarily on NPO
> While we focused on the limitations of NPO, we respectfully emphasize that NPO is not a narrow or fringe method, but rather the most widely adopted unlearning approach in existing LLM unlearning benchmarks, including TOFU and MUSE. Given NPO’s central role in the field, understanding and addressing its limitations is foundational to advancing the state of LLM unlearning. Our insights are original and, to the best of our knowledge, have not been previously explored in the unlearning literature.
>
> Beyond identifying this limitation, we introduce SimNPO and its both empirical and theoretical analyses, which have also been acknowledged by other reviewers. For example, **Reviewer JEx6** noted: “The paper … systematically validates SimNPO through theoretical derivations (e.g., gradient analysis in Appendix E) and extensive experiments (TOFU, MUSE, WMDP).” And **Reviewer s2gF** noted: “The paper provides sufficient empirical evidence that SimNPO is better than baselines considering the unlearning-utility trade-off in LLM unlearning.”
>
>
>
> # 2. Response to the potential negative impact of removing the reference model
>
> Thank you for this insightful question. We would like to clarify that within the framework of SimNPO, we did not observe any clear negative impact on either unlearning effectiveness or utility resulting from the removal of the reference model. Additionally, SimNPO benefits from computational and memory efficiency, as it eliminates the need to query and store a reference model during optimization. We further conducted a comparison of the computational efficiency among Retrain, NPO, and SimNPO. As shown in **Table R4**, NPO introduces additional overhead during the unlearning process for two reasons: (1) it requires storing an extra reference model, which increases GPU memory usage; and (2) it must compute $\log(\pi_{\text{ref}}(y \mid x))$ at each training step, leading to increased computational time.
>
> Table R4: Comparison of GPU memory usage and training time for Retrain, NPO, and SimNPO on the TOFU Forget05.
>
> | Method   | Memory (GB) | Time (min) |
> |:--------:|:-----------:|:----------:|
> | Retrain  |     20      |    120     |
> | NPO      |     27      |     36     |
> | SimNPO   |     21      |     25     |
>
> However, it is important to emphasize that this improvement does not come from simply removing the reference model from NPO. As noted in Lines 234–237, a naive removal of the reference model from Eq. (3) in NPO reduces it to a GA (gradient ascent), which disrupts the weight smoothing effect and degrades unlearning-utility performance.
>
> In terms of utility, the removal of the reference model in SimNPO does not lead to negative effects; on the contrary, it often results in better utility preservation compared to NPO. This improvement stems from SimNPO’s more effective gradient smoothing mechanism, which avoids the instability observed in NPO during the early stages of unlearning. This point is discussed in Line 219 (“Lack of gradient weight smoothing in the early stages of unlearning”) and supported by Figure 4.
>
>
> # 3. Response to the existence of forgotten performance bias in NPO and under what conditions does this bias arise
> Thank you for this insightful question. The reference model bias in NPO arises when the reference model misguides the unlearning process by assigning inconsistent gradient weights to forget samples of varying difficulty.
>
> To elaborate, let us revisit the reference-dependent weighting term in NPO’s gradient (Eq. 3): $w_{\theta}(x, y) = \frac{2 \, \pi_\theta(y \mid x)^\beta}{\pi_\theta(y \mid x)^\beta + \pi_{\text{ref}}(y \mid x)^\beta}$. This weighting scheme causes NPO to assign smaller gradient magnitudes to samples with higher output probabilities under the reference model, $\pi_{\text{ref}}(y \mid x)$. However, such samples are actually harder to forget than those with lower $\pi_{\text{ref}}$. As a result, if $\pi_{\text{ref}}$ varies across the forget set in a way that does not faithfully reflect the true unlearning difficulty of each sample, this leads to **uneven gradient assignment** and introduces **bias into the forgetting process**.
>
> This bias is particularly evident when forget examples vary in memorization level (Example 1) or response length (Example 2). In Example 1, NPO performs poorly on strongly memorized (harder) samples–characterized by higher $\pi_{\text{ref}}$--yielding a forget quality of 0.00, while achieving a forget quality of 0.47 on weakly memorized (easier) samples, as shown in Figure 1(b). A similar trend is observed in Example 2 for longer-response forgetting data vs. shorter data (Figure 2).
>
> # 4. Response to new bias from length normalization in SimNPO
>
> Thank you for your insightful and thought-provoking question.
>
> Using length normalization in SimNPO does **not introduce** bias with respect to sample length; rather, it **mitigates** such bias in NPO from forgetting data with different lengths. As explained in Response 3,  short data samples tend to have **higher $\pi_{\text{ref}}$**, they are generally **harder to unlearn**. In the SimNPO adaptive weight formulation: $w_{\theta}(x, y) = \frac{2 \, \pi_\theta(y \mid x)^\beta}{\pi_\theta(y \mid x)^\beta + 1} \cdot \frac{1}{|y|}$ this weight assignment naturally gives **larger gradient weights  to shorter samples**, thereby compensating for their higher unlearning difficulty. As shown in **Figure 2**, the forget quality of **NPO** on short vs. long samples is **0.47 vs. 0.81**, indicating substantial bias. In contrast, **SimNPO** achieves **0.91 vs. 0.97**, significantly reducing this gap and demonstrating better balance across sample lengths.
>
> To further validate the usefulness of length normalization, we conduct additional experiments in **Appendix F**. When the parameter $\beta \rightarrow 0$, the SimNPO gradient degenerates into **length-normalized weighted GradDiff**, whereas the NPO gradient degenerates into **standard GradDiff**. It can be seen from Figure A3 that although not outperforming the original NPO or SimNPO,  the forget quality of weighted GradDiff (0.1) is **substantially higher** than that of GradDiff (1e-9), highlighting the usefulness of length normalization.

---

### Official Review · Reviewer_eoQN · 2025-07-03

**Clarity:** 2
**Significance:** 2
**Originality:** 2
**Rating:** 4
**Confidence:** 3

**Summary:**

This paper identifies an issue in Negative Preference Optimization (NPO): reference model bias. The paper claims that this bias leads to uneven allocation of optimization effort across different data and ineffective gradient weight smoothing during the early stages of training. Inspired by SimPO, this paper proposes SimNPO to address this problem and mitigate the negative effects introduced by the reference model.

**Questions:**

See above

**Ethical Concerns:**

["NO or VERY MINOR ethics concerns only"]

**Final Justification:**

After the authors' clarifications, most of my concerns have been resolved, except that I still believe more series and sizes of base models should be included in the experiments. I set my final score to 4.

**Limitations:**

yes

**Quality:**

3

**Strengths And Weaknesses:**

## Strengths

- The SimNPO method proposed in this paper is clearly presented.
- Sections 3 and 4 provide useful insights through their analysis of the NPO objective and its gradient. Although I have some reservations about parts of the analysis, such detailed investigation contributes to the value of this work.
- Experiments on TOFU, MUSE, and WMDP validate the effectiveness of the proposed method.

## Weaknesses

- I have some reservations about the analysis presented in Section 4:
1. For example, in Example 1 on L1, I can understand that data with higher output probabilities under the reference model are harder to forget than data with lower output probabilities. However, according to Eq. (3), the latter would simply be forgotten at a slower rate, but should ultimately still be forgotten if the optimization is effective. Therefore, the forgetting performance of NPO on the forget data should not be significantly worse than that of SimNPO. This makes it unclear to me why, as shown in Figure 3, the forget quality of NPO is almost only half that of SimNPO.
2. In Example 2 on L1, if the reference model tends to produce longer yet lower-quality responses, then based on the analysis in Example 1, data with similar characteristics (i.e., longer and lower-quality) should be harder to forget. Why, then, is unlearning for shorter forget data more ineffective? This seems to contradict the analysis from Example 1.
3. According to the analysis on L2, NPO should initially forget the forget data at a relatively fast rate. Why, then, does the NPO forget quality curve in Figure 3 consistently remain below that of SimNPO throughout training?
- SimNPO leverages the idea of SimPO to remove the explicit influence of the reference policy. However, in my view, the effectiveness of this method compared to NPO depends heavily on the initial model itself. The experiments should therefore include more types of initial models (e.g., the Qwen series) and discuss how the choice of base model influences the performance of SimNPO.

If I have misunderstood any aspect of this paper, please feel free to correct me and discuss it with me. I would be open to adjust my score.

---

> ### Author Rebuttal · Authors · 2025-07-30
>
> We greatly appreciate Reviewer eoQN's insightful and constructive comments. Below, we provide detailed responses to address each of the points raised.
>
> # 1. Response to NPO vs. SimNPO in unlearning speed and the forget quality gap in Fig. 3
>
> The reviewer raises a concern that although NPO would be forgotten at a slower rate, it should ultimately still be forgotten if the optimization is effective. Therefore, the forgetting performance of NPO on the forgetting data should not be significantly worse than that of SimNPO. Yet, as shown in Figure 3, the forget quality of NPO is almost only half that of SimNPO.
>
> To address the above concern, we believe there may be a misunderstanding regarding the interpretation of forget quality (as used in TOFU; Figure 3). **Forget quality in TOFU is not a measure of unlearning speed**. Instead, it quantifies how closely the output distribution of an approximately unlearned model aligns with that of the Retrain model (the gold standard for exact unlearning) on the forget set (Line 171). A higher forget quality (closer to 1) reflects more faithful emulation of exact unlearning behavior, not merely faster forgetting.
>
> Based on the above explanation, the observation that NPO’s forget quality is nearly half that of SimNPO suggests that NPO is substantially less effective than SimNPO in approximating the desired unlearning behavior established by Retrain. The better unlearning dynamics of SimNPO over NPO can be attributed, in part, to the adaptive weighting term preceding the NPO gradient in Eq. (3): $
> w_{\boldsymbol{\theta}}(x, y) = \frac{2 \, \pi_\theta(y \mid x)^\beta}{\pi_\theta(y \mid x)^\beta + \pi_{\text{ref}}(y \mid x)^\beta}
> $.
>
> As we can see, NPO assigns smaller gradients to data points with higher $\pi_{\text{ref}}(y \mid x)$. However, as you rightly pointed out, data with higher $\pi_{\text{ref}}$ are actually harder to forget than those with lower reference probabilities. As a result, NPO tends to focus on forgetting samples with low $\pi_{\text{ref}}$, while neglecting samples with high $\pi_{\text{ref}}$, leading to what we define as reference model bias.
>
> This bias leads to high variance in NPO’s unlearning effectiveness across data points with differing levels of difficulty, as demonstrated by data memorization levels (Example 1, Line 199) and response lengths (Example 2, Line 208). While the reviewer suggested that NPO should eventually succeed in forgetting if the optimization converges, we respectfully argue that convergence alone is not sufficient to guarantee solution accuracy for non-convex optimization; therefore, a converged solution may still fall short of the desired forgetting outcome. This is empirically supported by Figure 1(b), where NPO fails to forget strongly memorized data (forget quality = 0), whereas SimNPO achieves a forget quality of 0.70 on the same subset.
>
>
> # 2. Response to unlearning  on shorter and longer samples
>
> There may be a misunderstanding regarding our statement that the “reference model tends to produce longer yet lower-quality responses.” We apologize for any confusion and offer the following clarification.
>
> By “low-quality,” we specifically refer to low-confidence generations, as indicated by **low output probabilities**. This phenomenon where longer responses tend to be assigned lower probabilities, was also observed in the SimPO paper. It was shown that generation length often counteracts the log-likelihood. As you correctly interpreted, data with higher output probabilities under the reference model are harder to forget than data with lower output probabilities.
>
> Therefore, unlearning on longer (lower-probability) data is easier compared to shorter data. This trend is empirically supported in Figure 2, where NPO performs significantly worse on shorter (harder-to-unlearn) samples compared to longer (easier-to-unlearn) ones (0.58 vs. 0.81 in forget quality). A similar pattern is observed in Figure 1(b): NPO shows poor performance on strongly memorized (harder) samples (forget quality = 0.00), while performing better on weakly memorized (easier) ones (0.47).
>
> # 3. Response to the unlearning speed between NPO and SimNPO
> Thank you for raising this question. As we responded to Question 1, interpreting the forget quality gap in Figure 3 as a measure of unlearning speed is inaccurate. Forget quality specifically measures how closely the output distribution of the unlearned model matches that of the Retrain model on the forget set. See more details in Response 1.
>
> If we consider convergence speed in terms of unlearning optimization iterations (using big-O notation), our theoretical analysis in **Appendix G shows that SimNPO and NPO share the same asymptotic convergence rate**. However, we emphasize that the advantage of SimNPO lies not in faster convergence per se, but in its improved unlearning dynamics. Specifically, SimNPO avoids the reference model bias inherent in NPO, which disproportionately affects forget samples of varying difficulty (e.g., differences in memorization levels or response lengths). Moreover, SimNPO introduces a more stable and effective weight smoothing scheme (Figure 4) over iterations to alleviate early utility drop and yield more balanced unlearning behavior (see further analyses via a mixture of Markov chains).
>
> # 4. Response to generality of SimNPO across different initial models
> Thank you for your suggestion. We selected a broad range of models, as outlined in **Table A1**, such as GPT-2 (referenced in Line 367, Page 7), LLaMA-2-chat 7B, LLaMA-2 7B, ICLM-7B, and Zephyr-7B-beta.
> To further validate the effectiveness of our method, we conducted **additional experiments** on WMDP with **Qwen2.5-14B** based on your suggestion. Here are the detailed experimental settings:
>
> - Dataset: WMDP, Composed of hazardous knowledge from bio/cybersecurity domains
> - Models: Qwen2.5-14B
> - Unlearning methods: NPO and SimNPO
> - Evaluation metrics: WMDP accuracy (to measure unlearning efficacy, lower is better) and MMLU accuracy (to measure utility preservation, higher is better)
>
> The results are shown in **Table R3**. As observed, SimNPO achieves a lower WMDP accuracy than NPO under the same MMLU, further demonstrating the effectiveness of our method.
>
> Table R3. Performance of Qwen2.5-14B on WMDP
> |        | WMDP | MMLU |
> |:------:|:-----:|:----:|
> | Origin |  0.81 | 0.77 |
> | NPO    |  0.35 | 0.69 |
> | SimNPO |  0.29 | 0.69 |

---

> > ### Comment · Reviewer_eoQN · 2025-08-06
> >
> > Thanks for the reply. I still believe that more base models should be included in the experiments. However, most of my concerns have been addressed, so I have decided to raise my score to 4.

---

> ### Author Response · Authors · 2025-08-06
> **Thank you for your recognition and score update**
>
> Dear eoQN,
>
> Thank you sincerely for raising your score and for recognizing our rebuttal efforts. We greatly appreciate your thoughtful feedback throughout the review process and are glad that most of your concerns have been resolved.
>
> Regarding the consideration of more base models, we will work to include additional evaluations in the revision. In the meantime, we would like to remark that we selected different base models as recommended for each unlearning benchmark (see Table A1). In our synthetic experiment, we also used GPT-2 to validate the Markov Chain analysis. Furthermore, during the rebuttal, we added results for Qwen2.5-14B on WMDP (Table R3). In the revision, we will explore more base models used in unlearning benchmarks and add an ablation study on base model choice.
>
> Thank you again for your diligent review and valuable comments.
>
> Authors

---

### Official Review · Reviewer_JEx6 · 2025-07-03

**Clarity:** 3
**Significance:** 3
**Originality:** 2
**Rating:** 4
**Confidence:** 3

**Summary:**

The paper identifies a critical limitation in the state-of-the-art Negative Preference Optimization (NPO) method, termed reference model bias, which arises from NPO's reliance on the reference model to guide unlearning, and then propose SimNPO (Simple NPO), a reference-free optimization framework inspired by simple preference optimization. SimNPO replaces NPO's reference-dependent reward formulation with a length-normalized, reference-free approach, positioning SimNPO as a significant advancement in LLM unlearning.

**Questions:**

How does SimNPO compare to non-preference-based unlearning methods (e.g., adversarial unlearning [28], or task arithmetic [53]) in terms of computational cost and scalability?

**Ethical Concerns:**

["NO or VERY MINOR ethics concerns only"]

**Limitations:**

yes

**Quality:**

3

**Strengths And Weaknesses:**

Strengths
- The paper provides a thorough analysis of NPO’s limitations (reference model bias) and systematically validates SimNPO through theoretical derivations (e.g., gradient analysis in Appendix E) and extensive experiments (TOFU, MUSE, WMDP).
- SimNPO outperforms NPO and baselines across multiple tasks (Tables 1–2, Fig. 6), suggesting broad applicability.

Weaknesses
- While SimNPO advances NPO, the core idea (reference-free optimization) is borrowed from SimPO. The adaptation to unlearning is novel but not groundbreaking.
- More results on general benchmarks like MMLU, AIME, HumanEval is welcome to demonstrate the universality of the method

---

> ### Author Rebuttal · Authors · 2025-07-30
>
> We sincerely thank Reviewer JEx6 for the thoughtful and constructive feedback. Below, we provide detailed responses to address each of the key questions raised.
>
> # 1. Response to novelty beyond SimPO
>
> While SimNPO may appear to be a simple methodological modification to SimPO, we respectfully emphasize that our work represents a novel contribution tailored to the domain of **LLM unlearning**, a substantially different problem space than the context originally considered in SimPO. Specifically, SimNPO was motivated, analyzed, and rigorously validated within the unlearning context: Our work fundamentally rethinks the design of unlearning objectives by identifying and overcoming the reference model bias in NPO.
>
> In particular, we are the first to (1) uncover the disproportionate gradient influence imposed by the reference model across forgetting data of varying difficulty (as exemplified via memorization level and response length), and (2) empirically and theoretically characterize how this bias leads to suboptimal forgetting outcomes. These findings have not been addressed in prior work. Our contributions go beyond methodological substitution. We provide a systematic investigation, including theoretical analysis via a Markov chain mixture model, empirical validation across multiple benchmarks, and robustness studies, as summarized in the “strength” session, “systematically validates SimNPO through theoretical derivations and extensive experiments.”
>
> We are also encouraged that other reviewers recognized our novelty. For example, **Reviewer eoQN** noted: “Sections 3 and 4 provide useful insights through their analysis of the NPO objective and its gradient.” **Reviewer BfAc** noted: “The paper thoroughly examines the bias limitations of the NPO method and provides a comprehensive analysis of the advantages of the proposed SimNPO. This solidifies the technical claims and offers valuable insights for the related communities into more fair unlearning mechanisms.” **Reviewer s2gF** noted: “The paper provides sufficient empirical evidence that SimNPO is better than baselines considering the unlearning-utility trade-off in LLM unlearning.”
>
>
> # 2. Response to the choice of benchmark
> Thank you for your valuable feedback.
>
> First, we would like to clarify that our current evaluations strictly follow the **official evaluation protocols** for each benchmark, as summarized in Table A1.
>
> Second, we conduct additional utility evaluations of our unlearned WMDP models on MathQA, which includes a wide range of mathematical reasoning types, including geometry, algebra, unit conversion, and multi-digit arithmetic (e.g., addition and subtraction involving 2–5 digit numbers). We further assess utility on the TruthfulQA benchmark, which measures factual consistency and truthfulness in LLM outputs.
>
> As shown in **Table R1**, SimNPO consistently outperforms NPO across all utility benchmarks, indicating a stronger retention of both general reasoning and factual knowledge.
>
> Table R1. Utility evaluation of the original model, NPO, and SimNPO unlearned WMDP models (Zephyr-7b-beta) on four downstream benchmarks: Addition, Subtraction, MathQA, and TruthfulQA. Unlearning efficacy is also reported, measured as 1 - Accuracy on the WMDP evaluation set.
>
> | Method   | Unlearning Efficacy | MMLU  | Addition | Subtraction | MathQA | TruthfulQA |
> |:--------:|:-------------------:|:-----:|:--------:|:--------:|:------:|:----------:|
> | Original   |        0.35         | 0.59  |  0.89    |  0.88    | 0.27   |   0.41     |
> | NPO      |        0.74         | 0.44  |  0.80    |  0.71    | 0.18   |   0.25     |
> | SimNPO   |        0.74         | 0.48  |  0.88    |  0.88    | 0.26   |   0.36     |
>
> # 3. Response to comparison with non-preference-based unlearning methods (e.g., adversarial unlearning, or task arithmetic) in computation cost.
> Regarding adversarial unlearning [1,2], it is typically implemented via min-max optimization, which involves not only standard forward and backward passes, but also additional perturbations on activations [1] or weights [2], followed by extra forward and backward passes. As a result, its computational cost is higher that of SimNPO.
>
> As for task vector, we evaluated its performance on the MUSE benchmark, and as shown in Table 2, its unlearning ability is very limited (almost completely ineffective). For example, on MUSE-Books, we use the VerbMem on $D_f$ metric (where lower is better) to measure residual memorization of the forget set. The original model yields a VerbMem score of 99.56, and task vector only slightly reduces this to 99.31. In contrast, SimNPO reduces the score to 0, indicating more effective forgetting.  Given that the task vector cannot achieve the effectiveness of unlearning, its computational cost might not be meaningful to compare in this context.
>
> In addition, task vector also involves model fine-tuning to obtain the task-specific model before computing the difference with the original model weights to form the vector; a process already employed in the MUSE benchmark. Furthermore, it is **less memory-efficient than SimNPO**, as it requires storing both the original and task-specific models prior to the task negation step.
>
> We further conducted a comparison of the computational efficiency among Retrain, NPO, and SimNPO. As shown in **Table R2**, NPO introduces additional overhead during the unlearning process for two reasons: (1) it requires storing an extra reference model, which increases GPU memory usage; and (2) it must compute $\log(\pi_{\text{ref}}(y \mid x))$ at each training step, leading to increased computational time.
>
> Table R2: Comparison of GPU memory usage and training time for Retrain, NPO, and SimNPO on the TOFU Forget05.
>
> | Method   | Memory (GB) | Time (min) |
> |:--------:|:-----------:|:----------:|
> | Retrain  |     20      |    120     |
> | NPO      |     27      |     36     |
> | SimNPO   |     21      |     25     |
>
> >[1] Sheshadri, Abhay, et al. "Latent adversarial training improves robustness to persistent harmful behaviors in LLMs." arXiv preprint arXiv:2407.15549 (2024).
> >
> >[2] Fan, Chongyu, et al. "Towards LLM unlearning resilient to relearning attacks: A sharpness-aware minimization perspective and beyond." arXiv preprint arXiv:2502.05374 (2025).

---

### Comment · Area_Chair_RxB1 · 2025-08-05
**Please read the authors’ rebuttal and join the discussion**

Dear Reviewers,

Thank you for your valuable contributions.

The authors have provided rebuttal to your reviews. Please carefully read their responses  as soon as possible, and indicate whether your concerns have been addressed. You are also encouraged to engage in discussion with fellow reviewers.

Note that simply **submitting "Mandatory Acknowledgement" without posting any feedback is NOT allowed**. *Let's be the kind of reviewers we’d appreciate for our own submissions*.

Best,

AC

---

### Decision · Program_Chairs · 2025-09-17

**Decision:**

Accept (poster)

**Comment:**

This paper revisits Negative Preference Optimization in the context of LLM unlearning, showing that simpler strategies can be surprisingly effective compared to NPO. The authors provide systematic analysis, clear methodology, and empirical evidence across multiple unlearning benchmarks. Reviewers found the paper well written and the empirical results convincing, though they also noted that the contribution is incremental, focusing on re-evaluating design choices rather than introducing fundamentally new techniques. The rebuttal further clarified the experimental setup and strengthened the empirical case. Overall, the paper is solid, clear, and practically useful, though the contribution is not groundbreaking. I recommend acceptance.